**Data Availability Statement:** All relevant data are within the paper and its Supporting Information files.

# Temporal changes in spike IgG levels after two doses of BNT162b2 vaccine in Japanese healthcare workers: Do spike IgG levels at 3 months predict levels 6 or 8 months after vaccination?

**Masaaki Takeuchi**[ORCID]*, **Akina Esaki, Yukie Higa, Akemi Nakazono**

Department of Laboratory and Transfusion Medicine, Hospital of University of Occupational and Environmental Health, School of Medicine, Kitakyushu, Japan

* takeuchi@med.uoeh-u.ac.jp

## Abstract

### Background

Accurate timing of BNT162b2 boosters to prevent breakthrough infections of coronavirus disease 2019 (COVID-19) requires reliable estimates of immune status. We hypothesized that spike IgG levels at 3 months after two doses of the BNT162b2 vaccine might predict subsequent spike IgG levels.

### Methods and results

Spike IgG levels were tested at 3, 6, and 8 months after the second dose of the BNT162b2 vaccine in 251 Japanese health care workers (median age: 39 years, female: 187).

The median level of spike IgG was 2,882 AU/mL at 3 months. This decreased to 875 AU/mL at 6 months and 579 AU/mL at 8 months. There were good correlations of log-transformed spike IgG levels between 3 and 6 months (r = 0.86) and between 3 and 8 months (r = 0.82). The correlation further improved after excluding three subjects who had possible COVID-19 infections (r = 0.91, r = 0.86). Log-transformed spike IgG levels at 6 or 8 months yields the following equation:

log spike IgG at 6 (8) months = 0.92 (0.86) X log spike IgG at 3 months– 0.23 (0.18). Predicted spike IgG at 6 months of $\geq$ 300 or < 300 AU/mL had 98% sensitivity, 47% specificity, and 94% accuracy for discriminating subjects whose actual spike IgG titers at 6 months were above or below 300 AU/mL. Corresponding values of predicted spike IgG at 8 months were 97%, 70%, and 93%, respectively.

### Conclusions

We conclude that predictive formulae using spike IgG levels at 3 months after two-dose vaccination with BNT162b2 reliably estimate subsequent spike IgG levels up to 8 months and provide useful information in terms of vaccination booster timing.

**Funding:** The authors received no specific funding for this work.

**Competing interests:** MT received a research grant from Abbott. Other authors declare that they have no conflicts of interest. However, this does not alter our adherence to PLOS ONE policies on sharing data and materials.

## Introduction

Among several coronavirus disease 2019 (COVID-19) vaccines, a lipid nanoparticle-formulated, nucleoside-modified messenger RNA encoding the severe acute respiratory syndrome coronavirus 2 (SARS-CoV-2) full length spike glycoprotein in a prefusion stabilized conformation, BNT162b2 (Pfizer/BioNTech), is one of the most widely used vaccines to prevent COVID-19 disease, and its efficacy was initially reported to be 91.3% up to 6 months after full-vaccination [1]. However, acquired immunity after vaccination does not persist at high levels, but begins to wane, and breakthrough infections frequently develop even in people who have received the requisite two dose series of BNT162b2 [2, 3]. Although new variants of concern that escape the immune system are another potential cause of lower vaccine effectiveness, serial reductions of both spike IgG and neutralizing antibodies have been reported in various publications [4–16], and these findings support a third vaccine dose to boost antibody production, to control resurgent COVID-19 [17–22]. Due to limited vaccine supply, an individualized booster vaccine schedule is recommended to conserve vaccine doses and to prioritize them to high-risk patients or developing countries. In our previous study, spike IgG levels two-weeks after the second dose of BNT162b2 ranged from 2,826 to 70,272 AU/mL in 67 COVID-19 naïve healthy Japanese healthcare workers [23], meaning that not all subjects may need a third vaccine dose within a fixed time window. A recent position paper regarding cancer patients recommends measuring spike IgG at 3 months in responder patients (spike IgG 3 to 4 weeks after second dose vaccination > 1,800 AU/mL) [24]. We hypothesized that spike IgG level at 3 month is a good time to assess the kinetics of waning immunity because all vaccinees have already reached peak levels of spike IgG [4, 13, 25–27], and some studies found a decline in antibody levels at 3 months [4, 13, 14, 26, 28]. Accordingly, the aims of this study were 1) to investigate temporal changes in spike IgG levels after two-dose vaccination with BNT162b2, and 2) to determine whether spike IgG levels at 3 months can predict spike IgG levels 6 or 8 months after two-dose vaccination.

## Materials and methods

### Study participants

This was a prospective, longitudinal, observational study in a single center. The study was approved by the institutional review board of the University of Occupational and Environmental Health, School of Medicine (approval number: UOEHCRB21-050). We advertised for study participants working in the university hospital (healthcare workers; HCWs). We also invited subjects who participated in the previous study [23]. Inclusion criteria were subjects who had received the first dose of BNT162b2 mRNA Covid-19 vaccine from March 22 to March 26, 2021 and the second dose from April 12 to April 23, 2021. Study participation was voluntary. Participant recruitment commenced on June 14, 2021 and ended on July 30, 2021. Written informed consent was obtained from all subjects who agreed to participate in the study.

### Antibody test

Blood samples were obtained from participants 3 months (median: 92 days [interquartile range; IQR: 91 to 97 days]) and 6 months (median: 190 days [IQR: 188 to 194 days]) after the second vaccination. During the study, the Japanese government decided to distribute a third dose of mRNA vaccine (BNT162b2) to HCWs starting in December 2021. Since the hospital chairman decided to administer a third vaccine dose to hospital HCWs from December 21 to December 24, a third blood sample was taken before the third vaccine dose was given, which

was equivalent to 8 months after the second vaccination (median: 243 days [IQR: 238 to 246 days] after the second dose). The titer of IgG against S protein's receptor biding domain (RBD) in sera from blood sample was measured by an ARCHITECT SARS-CoV-2 IgG II Quant assay on Architect system (Abbott Laboratories, Abbott Park, IL, USA). According to the instruction of IgG (RBD) assay, the cut-off index is 50.0 arbitrary units/mL (AU/mL) (https://www.fda.gov/media/146371/download) [29]. To exclude the possibility of previous SARS-CoV-2 infection, we also measured IgG antibodies against the nucleocapsid protein of SARS-CoV-2 (Abbott Diagnostics) at 3, 6, and 8 months in all subjects. IgG against nucleocapsid protein (N) of SARS-CoV-2 in sera was measured by an ARCHITECT SARS-CoV-2 IgG assay on Architect system (Abbott Laboratories). The presence or absence of nucleocapsid IgG antibodies to SARS-CoV-2 was determined by comparing the chemiluminescent relative light unit in the reaction to the calibrator relative light unit, which is calculated as an index (S/C). According to the instruction of IgG (N) assay, the cut-off index is 1.40 S/C (https://www.fda.gov/media/137383/download).

## Statistical analysis

Levels of spike IgG were expressed as the median, IQR, and the geometric mean with 95% confidence intervals (CIs), calculated using Student's t test on log-transformed data. Other numerical data were presented as medians and IQRs. Categorical variables were expressed as numbers or percentages. Subjects were divided into two groups according to gender. For the main analysis, spike IgG levels were log-transformed due to skewness of the data distribution which was verified using the Shapiro-Wilk test and Q-Q plots. Pearson's correlation analysis was used to compare spike IgG levels among time points. To predict the level of log-transformed spike IgG 6 and 8 months after the second injection, univariate linear regression analysis was performed with log-transformed spike IgG values at 3 months as a covariate. We also performed multivariate linear regression analysis, including age, sex, log-transformed spike IgG at 3 months after the second injection, and incremental values were determined. Using predictive formulae, we calculated predicted spike IgG levels at 6 and 8 months in every participant. Although there were no definite cut-off values of spike IgG levels for adequate immunity, a large Israeli study proposed that subjects whose spike IgG levels were $\leq$ 300 AU/mL were more likely to become re-infected with COVID-19, than those whose antibody levels were > 300 AU/mL [10]. A position paper also addressed that spike IgG levels < 280–300 AU/mL for the Abbott IgG II assay were defined as non-responder patients with cancer [24]. Thus, we used a spike IgG of 300 AU/mL as a cut-off value. We determined the diagnostic accuracy of predicted spike IgG levels at 6 and 8 months obtained from formulae to stratify subjects whose spike IgG levels above or below 300 AU/mL at 6 and 8 months. We also performed ROC analysis to determine the best cut-off values. To evaluate the accuracy of predicted spike IgG at 6 and 8 months, we calculated variance explained by predictive models based on cross-validation (VEcv) [30] and Lin's concordance correlation coefficient [31]. A two-sided p-value < 0.05 was considered significant. All statistical analyses were conducted using R software version 4.1.2 (R foundation for Statistical Computing, Vienna).

## Results

Among 264 subjects who participated in this study, 13 subjects received one or two vaccinations outside the time-window for inclusion, leaving 251 (median age: 39 years, 187 were female) subjects in the study population. The vaccination interval was 3 weeks in 247 subjects (98%) and 4 weeks in 4 subjects (2%). Of 251 participants, 62 also participated in the previous study [23]; thus, they had spike IgG data just before their second injection and 2 weeks after

the second, allowing a comparison of spike IgG levels at five time points. Spike IgG levels 3 months after the second vaccination were available for all subjects. Spike IgG levels at 6 and 8 months were obtained in 250 and 249 subjects, respectively. Table 1 presents general characteristics of the study participants.

None had immunocompromised, and none had taken immunosuppressive medications. 73 HCWs (29%) were working in COVID-19 wards. Nucleocapsid IgG of > 1.40 S/C, which was suggestive previous COVID-19 infection, was observed two subjects.

## Spike IgG antibodies

The median, IQR, and geometric mean and its 95% CI 3 months after the second vaccination were 2,882 AU/mL, 1,790 – 3,998 AU/mL, 2,692 AU/mL, and 2,483– 2,917 AU/mL, respectively. Corresponding values at 6 months were 875 AU/mL, 590 – 1,359 AU/mL, 871 AU/mL, and 800 – 948 AU/mL, respectively. These values further decreased to 579 AU/mL, 384 – 914 AU/mL, 603 AU/mL, and 553 – 656 AU/mL 8 months after the second injection. However, none had a spike IgG < 50 AU/mL even at 8 months. Of 62 subjects with five spike IgG datasets, spike IgG showed the highest value 2 weeks after the second vaccination, followed by an approximately 82% reduction 3 months after the second vaccination (Fig 1).

Correlations of log-transformed spike IgG levels at three time points are shown in Fig 2. Although there were two outliers, good linear correlations were noted between 3 and 6 months (r = 0.86, p<0.001) and between 3 and 8 months (r = 0.82, p<0.001). The correlation was almost perfect between 6 and 8 months (r = 0.98, p<0.001). Of two outliers, one had symptomatic SARS-CoV-2 infection with positive results from a reverse transcriptase-polymerase chain reaction (PCR) at 5 months and nucleocapsid IgG of 3.47 S/C at 6 months after the second BNT162b2 injection. The other denied any symptoms related to SARS-CoV-2 infection, and nucleocapsid IgG was < 1.40 S/C at all three time points.

If we excluded the aforementioned two outliers and another subject whose nucleocapsid IgG at 3 months of 6.43 S/C, correlation coefficients further improved to 0.91 between 3 and 6

**Table 1. Clinical characteristics in study subjects.**

| Variable | |
|---|---|
| Age (years), median (IQR) | 39 (29, 47) |
| Male n (%) | 64 (25%) |
| Occupation | |
| Nurse | 145 (58%) |
| Technician | 65 (26%) |
| Doctor | 20 (8.0%) |
| Nutritionist | 7 (2.8%) |
| Pharmacist | 4 (1.6%) |
| Clerk | 4 (1.6%) |
| Others | 6 (2.4%) |
| Hypertension | 12 (4.8%) |
| Diabetes | 5 (2.0%) |
| Hyperlipidemia | 4 (1.6%) |
| Coronary artery disease | 0 (0%) |
| Chronic kidney disease | 0 (0%) |
| Immunocompromised disease | 0 (0%) |
| Subjects involve in the treatment of COVID patients | 73 (29%) |

Data are expressed as median (IQR) or number (%).

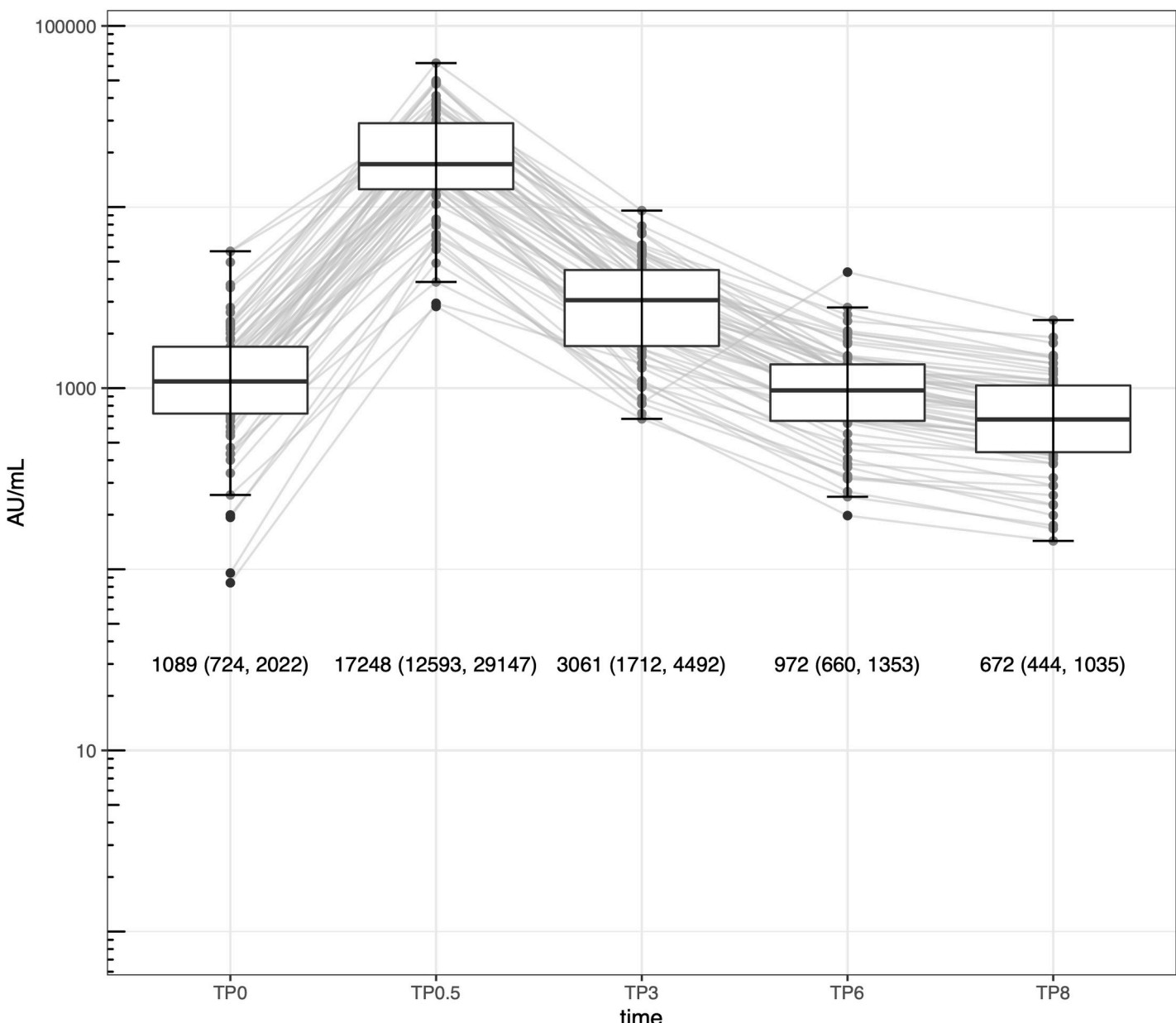

**Fig 1. Serial change in spike IgG levels after two-dose BNT162b2 vaccination in 62 subjects.** TP0, just before the second vaccination; TP0.5, 2 weeks after the second vaccination; TP3 (6, 8), 3 (6, 8) months after the second vaccination. Numerical values are expressed as medians (25th percentile– 75 th percentile).

months and 0.86 between 3 and 8 months. Since two had a definite breakthrough infection and the other was suspected of a potential infection, we excluded these three cases from linear regression analysis.

## Linear regression analysis

Univariate linear regression analysis revealed that log-transformed spike IgG levels at 3 months were significantly associated with log-transformed spike IgG levels at 6 months (Table 2).

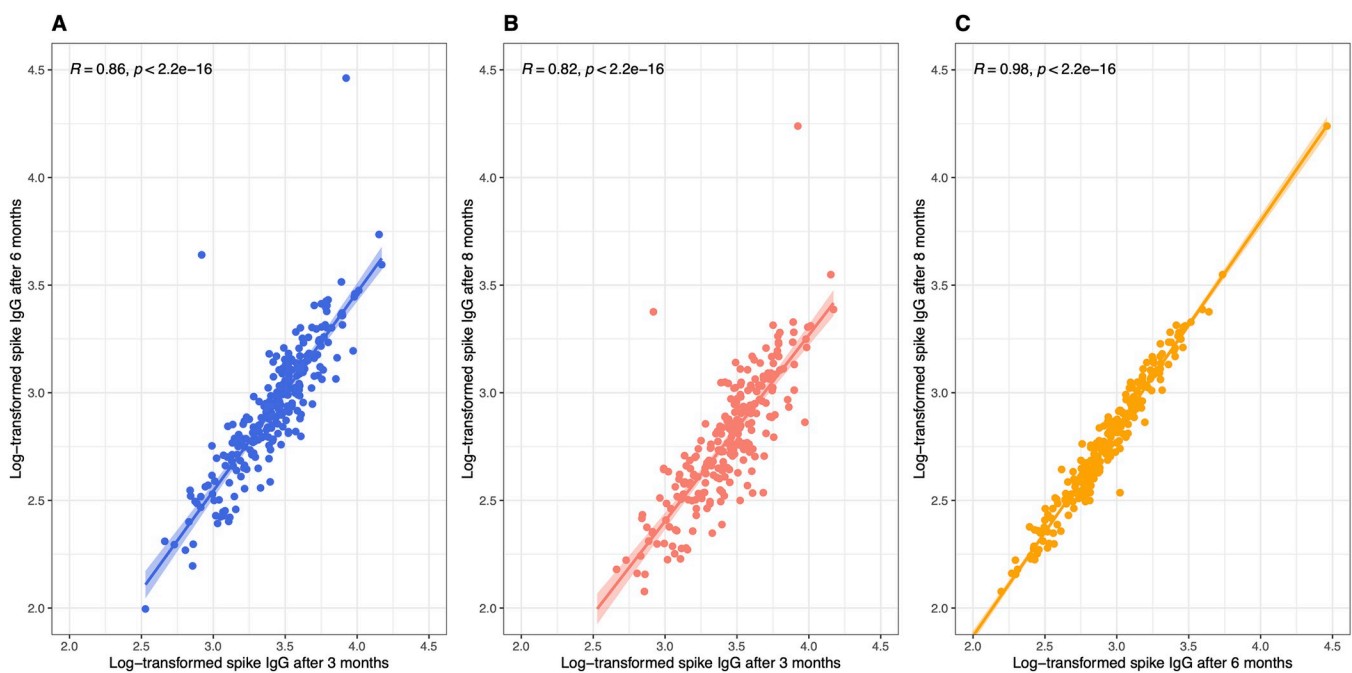

**Fig 2.** A linear correlation between spike IgG levels at 3 months and 6 months (panel A), at 8 months (panel B), and at 6 months and 8 months (panel C).

Multivariate analysis showed that log-transformed spike IgG at 3 months was significantly associated with log-transformed spike IgG at 6 months after adjusting for age and gender. There was also no incremental improvement from adding age and gender over log-transformed spike IgG at 3 months alone (p = 0.058). Log-transformed spike IgG at 6 months can be predicted as follows:

log spike IgG at 6 months = 0.92 X log spike IgG at 3 months– 0.23

Log-transformed spike IgG at 3 months was significantly associated with log-transformed spike IgG at 8 months with and without adjustments for age and gender (Table 2). There was a significant incremental value of adding age and gender over log-transformed spike IgG at 3 months alone (p = 0.027). Using log-transformed spike IgG at 3 months, corresponding values at 8 months can be predicted by the following equation:

log spike IgG at 8 months = 0.86 X log spike IgG at 3 months– 0.18

**Subgroup analysis according to gender.** Linear regression analysis was performed separately for gender (Fig 3). The regression slope did not differ between male and female, but the intercepts differed significantly in all comparisons.

**Table 2. Univariate and multivariate linear regression analysis for the association of log-transformed spike IgG titers at 6 months and that at 8 months.**

| type | | Predicting log spike IgG at 6 months | | | | Predicting log spike IgG at 8 months | | | |
|---|---|---|---|---|---|---|---|---|---|
| | coefficients | Beta coefficients | SE | t-value | p-value | Beta coefficients | SE | t-value | p-value |
| **univariate** | (intercept) | -0.228 | 0.091 | -2.50 | 0.013 | -0.181 | 0.112 | -1.60 | 0.11 |
| | log spike IgG at 3 months | 0.921 | 0.027 | 34.76 | <0.001 | 0.859 | 0.032 | 26.17 | <0.001 |
| **multivariate** | (intercept) | -0.210 | 0.105 | -2.00 | 0.047 | -0.172 | 0.130 | -1.32 | 0.187 |
| | log spike IgG at 3 months | 0.921 | 0.027 | 33.53 | <0.001 | 0.862 | 0.034 | 25.39 | <0.001 |
| | age | -0.001 | 0.001 | -1.023 | 0.308 | -0.001 | 0.001 | -0.93 | 0.352 |
| | female | 0.036 | 0.017 | 2.16 | 0.032 | 0.052 | 0.021 | 2.53 | 0.012 |

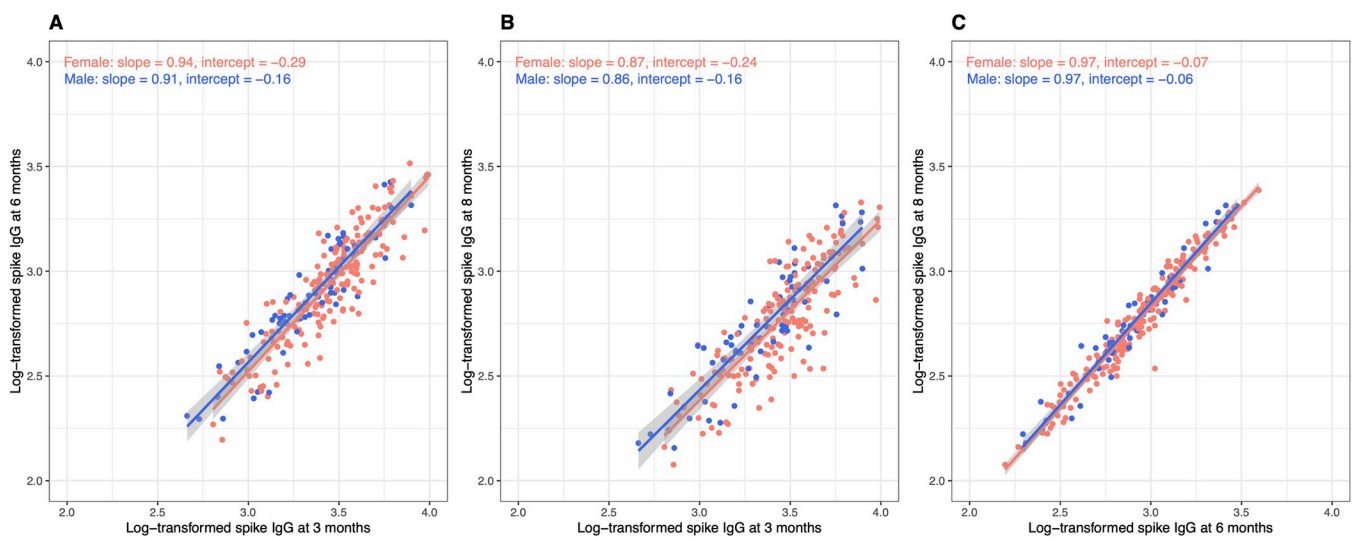

**Fig 3.** A linear regression analysis between spike IgG level at 3 and 6 months (panel A), at 8 months (panel B), and at 6 and 8 months (panel C), stratified by gender.

### Predictive accuracy of the model

We calculated predicted log transformed spike IgG levels at 6 months and 8 months, and back transformed to original values. Table 3 shows diagnostic accuracies of several cut-off values from equations for discriminating spike IgG of $\geq$ 300 AU/mL or < 300 AU/mL at 6 months and 8 months. Predicted spike IgG of $\geq$ 300 AU/mL or < 300 AU/mL at 6 months had 98% sensitivity, 47% specificity, and 94% accuracy for discriminating subjects whose observed spike IgG levels at 6 months were $\geq$ 300 AU/mL or < 300 AU/mL. Corresponding values of predicted spike IgG at 8 months were 97%, 70%, and 93%, respectively. ROC analysis revealed that the best cut-off value of predicted spike IgG levels at 6 months was 474 AU/mL with 89% sensitivity and 100% specificity. The corresponding best cut-off value of spike IgG at 8 month was 390 AU/mL with 87% sensitivity and 91% specificity.

S1 Fig shows correlation between observed spike IgG levels and predicted spike IgG levels at 6 months and 8 months. VEcv and Lin's CCC were 83.2% and 0.91 at 6 months. The corresponding values were 73.7% and 0.85 at 8 months after two-dose vaccination.

**Table 3. Predictive accuracy for spike IgG > 300 AU/mL at 6 months and 8 months.**

| 6 months | | | |
|---|---|---|---|
| Cut-off value using prediction (AU/mL) | Sensitivity | Specificity | Accuracy |
| 300 | 98% (225/230) | 47% (8/17) | 94% (233/247) |
| 350 | 95% (218/230) | 47% (8/17) | 91% (226/247) |
| 400 | 93% (215/230) | 76% (13/17) | 92% (228/247) |
| 450 | 90% (208/230) | 96% (16/17) | 91% (224/247) |
| 475 | 89% (204/230) | 100% (17/17) | 89% (221/247) |
| 500 | 87% (199/230) | 100% (17/17) | 87% (216/247) |
| 8 months | | | |
| Cut-off value using prediction (AU/mL) | Sensitivity | Specificity | Accuracy |
| 300 | 97% (206/213) | 70% (23/33) | 93% (229/246) |
| 350 | 92% (195/213) | 82% (27/33) | 90% (222/246) |
| 390 | 87% (185/213) | 91% (30/33) | 87% (215/246) |
| 400 | 85% (182/213) | 91% (30/33) | 86% (212/246) |

**Table 4. Estimated spike IgG levels post-vaccination.**

Spike IgG at 6 months

| spike IgG levels at 6 months (AU/mL) | 300 | 500 | 750 | 1,000 | 1,500 | 3,000 |
|---|---|---|---|---|---|---|
| spike IgG at 3M used for estimation (AU/mL) | 876 | 1,526 | 2,371 | 3,242 | 5,038 | 10,702 |
| Approximate spike IgG levels at 3M (AU/mL) | 900 | 1,500 | 2,400 | 3,200 | 5,000 | 10,000 |

Spike IgG at 8 months

| spike IgG levels at 8 months (AU/mL) | 300 | 500 | 750 | 1,000 | 1,500 | 3,000 |
|---|---|---|---|---|---|---|
| spike IgG at 3M used for estimation (AU/mL) | 1,229 | 2,226 | 3,567 | 4,985 | 7,987 | 17,884 |
| Approximate spike IgG levels at 3M (AU/mL) | 1,200 | 2,200 | 3,600 | 5,000 | 8,000 | 18,000 |

We used the following equation:

Log-transformed spike IgG at 3 months = (predicted log-transformed spike IgG at 6 months + 0.23) / 0.92

We used the following equation:

Log-transformed spike IgG at 3 months = (predicted log-transformed spike IgG at 8 months + 0.18) / 0.86

AU, arbitrary unit; M, months.

Using equations, we made a nomogram showing minimum spike IgG levels at 3 months in order to maintain certain levels of spike IgG at 6 and 8 months (Table 4).

Table 5 is another nomogram showing the month at which we expect to reach the cut-off level of spike IgG (300 AU/mL) after two-dose vaccination. For example, if spike IgG at 3 months was 800 AU/mL, expected elapsed time to reach spike IgG level < 300 AU/mL would be 5.5 months.

## Discussion

The main findings of this study are as follows: (1) spike IgG levels decreased from a median of 2,882 AU/mL, 3 months after the second injection to a median of 579 AU/mL after 8 months with wide individual variation; (2) spike IgG levels 6 and 8 months after the second vaccine dose were linearly correlated with those at 3 months; (3) Predicted spike IgG levels of 300 AU/mL estimated from spike IgG levels at 3 months accurately stratified subjects whose observed spike IgG levels were ≥ 300 AU/mL or < 300 AU/mL at 6 months with 94% accuracy and at 8 months with 93% accuracy.

### Previous studies

Recent breakthrough infections in countries where the two-dose vaccination rate is high, have raised concerns that immunity after the vaccination does not last long, clinically or serologically [2, 3]. Although an earlier study reported that the vaccine is highly effective against any

**Table 5. Estimated elapsed time when spike IgG level declines below 300 AU/Ml.**

| spike IgG levels at 3 months (AU/mL) | 500 | 600 | 800 | 1,000 | 1,200 | 1,500 | 2,000 | 3,000 |
|---|---|---|---|---|---|---|---|---|
| Predicted elapsed time when spike IgG levels becomes < 300 AU/mL (month) from 3 months after two-dose vaccination | 0 | 0.91 | 2.46 | 3.58 | 4.43 | 5.43 | 6.62 | 8.26 |
| Approximate elapsed time (month) after two-dose vaccination | 3 | 4 | 5.5 | 6.5 | 7.5 | 8.5 | 10 | 11 |

Values of the second raw of the table was calculated as follows.

From two formulae (log spike IgG at 6 months = 0.92 X log spike IgG at 3 months– 0.23, log spike IgG at 8 months = 0.86 X log spike IgG at 3 months– 0.18)

We supposed that the coefficient declined linearly according to month (M) (0.08/3 = 0.027, 0.14/5 = 0.028).

Thus, log spike IgG of 300 = (1–0.027 M) x log spike IgG at 3 months– 0.23.

Predicted elapsed month (M) = 37 x (1 –(2.71 / log spike IgG at 3 month))

grade of COVID-19 infection up to 6 months after two-dose vaccination [1], a recent study from Qatar reported that BNT162b2 vaccine effectiveness against *any* SARS-CoV-2 infection declined gradually after one month with the decline accelerating after the fourth month to reach 20%, 5 to 7 months after the second dose [32]. Similar results were obtained in the United States [33]. Another study demonstrated that the breakthrough infection rate was correlated with elapsed time after BNT162b2 vaccination [34].

Several studies have measured short- [26, 35] and long-term changes [4–16] in spike IgG and neutralizing antibody levels up to 9 months after two doses of BNT162b2, and all studies have revealed that antibody levels drop over time. Although some studies estimated antibody half-lives using either non-linear regression analysis [11] or one-compartment modeling [4], most estimates report mean values at each time point, and do not present individual values, which vary quite widely.

Although there is no definite cut-off value of spike IgG to prevent COVID-19 infection, Kertes et al. [10] determined the relationship between spike IgG levels and subsequent PCR-confirmed infection in a large number of Israeli subjects. The authors reported that the proportion of participants with positive PCR results was 1.2% for those with spike IgG below 150 AU/mL, 1.3% with spike IgG levels between150 AU/mL and 300 AU/mL, and 0.2% of those with spike IgG levels above 300 AU/mL. A recent position paper about cancer patients recommends a third vaccination in non-responder patients, defined as those whose spike IgG levels < 40 binding antibody units (BAU) / mL (< 280–300 AU/mL for the Abbott IgG II assay) [24]. Thus, we used a spike IgG cut-off value of 300 AU/mL as a reflection of a high risk of breakthrough infection and non-responder after the vaccination.

## Current study

Overall, median values of spike IgG 6 months after vaccination became one-third of levels at 3 months. Table 6 summarizes reported spike IgG levels at 3 and 6 months after two doses of the BNT162b2 vaccine.

IgG levels at 3 and 6 months varied widely among studies. The percent reduction of spike IgG levels from 3 months to 6 months ranged from 48% to 86%. However, it is interesting to note that the percent reduction of spike IgG levels in our study (70%) was quite similar to the corresponding value observed in three longitudinal studies (68%, 74%, 68%), which suggests that the decay kinetics of spike IgG after two-dose BNT162b2 vaccination may be relatively constant. Although the median of spike IgG levels at 3 month was 2,882 AU/mL, individual

**Table 6. Reported spike IgG levels at 3 months and 6 months after two doses of BNT162b2 vaccine.**

| Author | Study type | Subjects | Country | Spike IgG at 3M | Spike IgG at 6M | % reduction |
|---|---|---|---|---|---|---|
| Bayart | longitudinal | HCWs | Belgium | 6,050 AU/mL (n = 158) | 949 AU/mL (n = 158) | 68% |
| Naaber | longitudinal | HVs | Estonia | 5,226 AU/mL (n = 122) | 1,383 AU/mL (n = 122) | 74% |
| Rode | longitudinal | HCWs | Croatia | 2,977 AU/mL (n = 405*) | 966 AU/mL (n = 337*) | 68% |
| Guiomar | cross-sectional | HCWs | Portugal | 6,812 AU/mL (n = 32) | 1,070 AU/mL (n = 72) | 86% |
| Israel | cross-sectional | PB | Israel | 2,383 AU/mL (n = 200) | 765 AU/mL (n = 440) | 68% |
| Kertes | cross-sectional | PB | Israel | 2,706 AU/mL (n = 827) | 1,411 AU/mL (n = 1,820)** | 48% |
| Our study | longitudinal | HCWs | Japan | 2,882 AU/mL (n = 251) | 875 AU/mL (n = 250) | 70% |

Spike IgG levels are presented as mean or median.

HCWs, health care workers; HV, healthy volunteers; n, number; M, month; PB, population based.

*: exact number not available

**: ≥150 days

values ranged widely from 337 AU/mL to 14,837 AU/mL. Spike IgG levels at 6 months ranged even more widely from 99 AU/mL to 28,964 AU/mL due to a definite breakthrough infection in one case. Although there are different recommendations and guidelines about adjusting the period for introducing the third dose, based on specific characteristics of the population, especially in fragile, elderly, and multi-comorbid populations, an individualized vaccine schedule should be established to maintain effective prevention of breakthrough infection under limited vaccine supply. It would also reduce mass vaccination surges, which may reduce pressure on HCWs.

Since we obtained paired data on spike IgG in the same subjects during a relatively brief sampling time, we had an opportunity to study the temporal decline in spike IgG levels after two-dose vaccination of BNT162b2. We found that spike IgG levels at 6 and 8 months were highly correlated with those at 3 months. Multivariate regression analysis revealed that other parameters, such as age and gender did not greatly affect this association. Subgroup analysis showed similar regression line slopes between males and females. The result suggests that after reaching peak values, spike IgG levels decrease at a relatively constant rate, irrespective of age or gender. More importantly, our data indicate that spike IgG levels at a given sampling point are the main driver of subsequent spike IgG levels. Our previous study also supports this hypothesis [23].

Prediction of spike IgG of $\geq$ 300 AU/mL or $<$ 300 AU/mL at 6 months using the formula (Predicted log spike IgG at 6 months = 0.92 X log spike IgG at 3 months– 0.23) had a higher sensitivity (98%) but lower specificity (47%) for discriminating observed spike IgG of $\geq$ 300 AU/mL or $<$ 300 AU/mL at 6 months. If we used higher cut-off values (450 AU/mL), specificity (96%) increased with some loss of sensitivity (90%). The reason was because predictions were usually noisy. Quite often when observed values were relatively low, they were lower than predicted values, but when observed values were relatively high, they were higher than predicted values [30]. These trends are clearly shown in S1 Fig.

Our findings raise an important question of whether spike IgG levels 3 months after the second vaccine dose are optimal to predict future spike IgG levels. Although an analysis comprising only three sampling points does not answer this question definitively, we think that best time to predict future spike IgG levels may be from 1 to 3 months after the second vaccination. While reliability would probably improve further if we measured spike IgG levels closer to 6 months, this reduces the clinical utility of such measurements.

## Clinical implications

We demonstrated tight linear correlations between log-transformed spike IgG levels at 6 and 8 months with those at 3 months. Since addition of age and gender had no or little incremental value, a simple formula can be used to predict spike IgG levels at 6 or 8 months using only spike IgG levels at 3 months (Table 4). According to the table, spike IgG levels of 900 AU/mL represent approximately the minimum amount of spike IgG necessary at 3 months to maintain spike IgG of $\geq$ 300 AU/mL at 6 months. The table also provides minimal spike IgG levels at 3 months required to maintain spike IgG levels of 500, 750, 1,000, 1,500 and 3,000 AU/mL at 6 or 8 months. This table provides predictive spike IgG levels at 6 months in individual subjects, for whom we have spike IgG levels at 3 months after two-doses of BNT162b2 vaccine. Since spike IgG levels at 3 months vary widely among subjects, this information is quite useful for individualized third-dose vaccine schedules. Another nomogram (Table 5) shows at which month after vaccination one expects to reach the pre-defined spike IgG cut-off level (300 AU/mL) based on levels at 3 months. This nomogram provides information when spike IgG levels become $<$ 300 AU/mL in each subject who measured spike IgG levels at 3 months; thus, it supports individual scheduling for a third dose.

## Study limitations

This study has several limitations. First, the study population comprised healthy Japanese HCWs, and it may not be legitimate to extrapolate our results to other populations, such as elderly subjects ($> 65$ years), immunocompromised patients, or subjects who received other type of vaccines. Second, we could not determine spike IgG levels more than 8 months after the second dose because most participants had already received a third dose of BNT162b2 vaccine by that time. Third, although we generated simple equations, a good correlation between predicted and measured values was expected because we used the same dataset. A comparison of the equations obtained applying the same method described in this study to other cohorts will be useful to further support the accuracy of the predictions for healthy individuals of different ethnicities. Forth, the equation derived in this study will not be applicable to spike IgG dynamics after the third vaccination. However, it may be possible to establish a spike IgG threshold that ensures a statistical likelihood of immunity to serious COVID disease, hospitalization, and death. Further studies will be required to investigate whether similar correlation can be observed at other elapsed times after the third vaccination. Finally and more importantly, although spike IgG levels correlate with vaccine efficacy [36], we did not measure neutralizing antibody titers, which are better reflection of protective immune responses after vaccination. However, neutralizing antibody assays are costly and not fully automatic, resulting in low penetration in the general clinical field. They also require additional equipment. We believe that spike IgG monitoring could offer a relatively low-cost monitoring strategy in all situations [24].

## Conclusions

Spike IgG levels 6 and 8 months after two-dose vaccination with BNT162b2 are closely correlated with spike IgG levels at 3 months. Thus, spike IgG levels at 3 months predict subsequent spike IgG levels with high accuracy. Nomograms generated from results of this study provide useful information for individual scheduling for third vaccine dose.

## Supporting information

**S1 Fig.** A linear correlation between observed spike IgG level (x-axis) and predicted spike IgG level (y-axis) at 6 months (panel A), at 8 months (panel B).
(TIF)

**S1 Data.**
(XLSX)

## Acknowledgments

We thank all study participants.

## Author Contributions

**Conceptualization:** Masaaki Takeuchi.

**Data curation:** Akina Esaki, Yukie Higa.

**Formal analysis:** Masaaki Takeuchi.

**Methodology:** Masaaki Takeuchi, Akina Esaki, Yukie Higa, Akemi Nakazono.

**Writing – original draft:** Masaaki Takeuchi.

**Writing – review & editing:** Masaaki Takeuchi, Akina Esaki, Yukie Higa, Akemi Nakazono.

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
