## [Decision Letter · Decision Letter 0]

31 Mar 2022

PONE-D-22-01729Prediction of Spike IgG Levels in Japanese Healthcare Workers 6 and 8 Months after the Second Dose of the BNT162b2 VaccinePLOS ONE

Dear Dr. Takeuchi,

Thank you for submitting your manuscript to PLOS ONE. After careful consideration, we feel that it has merit but does not fully meet PLOS ONE’s publication criteria as it currently stands. Therefore, we invite you to submit a revised version of the manuscript that addresses the points raised during the review process. The methodology is important for PLOS ONE.

We look forward to receiving your revised manuscript.

Kind regards,

Etsuro Ito

Academic Editor

PLOS ONE

Journal Requirements:

“No.”

Reviewers' comments:

Reviewer's Responses to Questions

**Comments to the Author**

1. Is the manuscript technically sound, and do the data support the conclusions?

Reviewer #1: Yes

Reviewer #2: Yes

Reviewer #3: Partly

2. Has the statistical analysis been performed appropriately and rigorously? 

Reviewer #1: Yes

Reviewer #2: No

Reviewer #3: Yes

3. Have the authors made all data underlying the findings in their manuscript fully available?

Reviewer #1: No

Reviewer #2: Yes

Reviewer #3: Yes

4. Is the manuscript presented in an intelligible fashion and written in standard English?

Reviewer #1: Yes

Reviewer #2: Yes

Reviewer #3: No

5. Review Comments to the Author

Reviewer #1: The manuscript by Takeuchi and colleagues shows statistics-supported prediction of SARS-CoV-2 Spike IgG levels 6 and 8 months after two doses of a COVID-19 mRNA vaccine in a healthy population. Overall the paper is written well and provides a useful and easy method to predict individual evolution of antibody titers, with important clinical implications, in particular the possibility of individual scheduling of a third vaccine dose.

Major comments:

1. It’s not clear whether the enrolled participants were all naïve at the time of the vaccination or some of them had a SARS-CoV-2 infection before the first dose. This needs to be better explained as the two groups may have different starting titers and different decay kinetics. Does this also explain the great variability of sIgG2 titers shown in Figure 1? If numbers support a further sub-analysis, it would be interesting to assess the prediction in the two groups, separately.

2. Authors should better describe the rationale for performing the age sub-analysis defining 50 years as the cut-off for the comparison. Being the median age of 39, I would expect that there are not many participants aged greater than 50. How many healthcare workers belonged to the two groups? Would the results change if you define another age cut-off, such as the median age or 40 years?

3. Using a cut-off of 300 resulted in a very low specificity at 6 and 8 months, which may raise concern on the usefulness of performing such a test at these later timepoints. This should be commented in the discussion. Can you perform a ROC analysis to define a new cut-off, which increases specificity and does not lower sensitivity too much? How would this change your prediction nomograms?

4. Another type of nomogram could be more useful to show at which month after vaccination you expect to reach the cut-off level (here 300) based on titers measured at 3 months. This would better and more directly support an individual scheduling for a third dose.

5. Many studies have shown the decay of Spike-IgG titers after vaccination. Although beyond the scope of this study, a comparison of the equations obtained applying the same method described in this study to other published cohorts will be useful to further support the accuracy of the predictions for healthy individuals of different ethnicities.

Minor issues:

1. Check spelling of “Spike” throughout the text as it should be always capitalized.

2. The short terms sIgG1, sIgG2 and sIgG3 remind “serum IgG isotypes 1, 2 and 3”. I would change them into either TP1, T2, TP3… (timepoint) or even with the actual timing (3 mths, 6 mths, 8 mths). There’s no need to specify Spike IgG as it is clear from the y axis.

3. Please provide a Source Data file for each figure and table or a unique supplemental summary table with IgG titers for each participants at different time points and information about the age and gender category used for the sub-analyses.

Reviewer #2: Results of this study are consistent with current literature in the field. However, on the whole, the paper is not suitable for publication in its present form.

Specific Comments:

At the time of the study, booster (3rd dose) was already approved and in use in many countries. It is now common globally. Hence studying IgG levels at 8 months after the second dose make the significance of this study questionable. Also, it is important to perform measurements of neutralizing antibody titers, which are a better reflection of protective immune responses. Measuring total S-IgG levels after 3 months do not offer a complete picture of the virus neutralization capacity. And, total S-IgG levels cannot be the sole predictor for adequate immunity post-vaccination and is not sufficient to determine who needs a 3rd vaccine dose.

The authors need to be consistent using the terms SARS-CoV-2 and COVID-19.

The authors use the terms “injections, vaccination, and dose” and “spike IgG titers, values and levels” interchangeably. There needs to be a consistency in the use of specific terms.

Reviewer #3: A study by Takeuchi and colleagues explored temporal changes in the spike IgG titers after two doses of BNT162b2 vaccine and tried to determine whether spike IgG levels at 3 months can predict levels 6 or 8 months after full vaccination.

In general, this topic is highly relevant and the recruitment of study participants is done correctly. Surely, the high value of this study is the longitudinal collection of data during 3 (5) points in time, but the presentation of the results is pretty modest, with several critical segments that need to be improved before considering this paper for final publication. Several parts of the manuscript need to be rephrased.

I find the title of the paper a bit inappropriate since IgG levels are actuality measured at 6 and 8 months and not predicted. What authors did is that they derived formula for predicting values (using measured vales as dependent variables in the linear regression) so this is not main finding of the study and it might be misleading, please consider rephrasing it.

Point-by-point comments are listed below.

Introduction: this part needs further elaboration on the background of this topic, stronger rationale why this kind of study was needed.

line 46: “Although new variants of concern that escape the immune system are another potential cause of vaccine ineffectiveness…” please replace ineffectiveness with “lower effectiveness” since it’s incorrect.

Methods: why authors opted for measurement at 3 months? What is the rationale behind this decision?

Why there was a cut-off at 50 AU/mL? further, a reference would be helpful.

line 83: the authors state that IgG anti-N was measured “at least one time”, this is a potential issue and remains unclear since it may compromise the level of IgG measured? these should have been measured at all time points when IgG anti-S were measured? Without this how is that the authors may be confident that these IgG anti-S are the effect of the vaccine and not combined effect with (a)symptomatic COVID?

Statistical analyses: why was the data log-transformed? Skewed distribution? Any tests and visualization techniques used for the exploration of data normality (Q-Q plot, K-S test, Shapiro-Wilk test, etc.)?

line 91: the authors state that “Subjects were divided into two groups according to age (below 50 years or 50 years and above)”. What is the rationale behind this? Which hypothesis is tested there? This is much unclear, especially because a table with general description of the participants is missing.

Results: this section needs major improvements, it need to be rephrased and presented in more concise, straightforward and clear way.

A Table with the description of general characteristics of the study participants is highly needed here.

Also, there no mention of the potential comorbidities, medication used, vaccine reactogenicity, which all may have affected the IgG levels measured here?

On which department the study participants work on, are they involved in the treatment of the COVID patients? This might have potentially exposed them to the virus and booster their IgG levels, especially problematic if IgG anti-N were not measured for all participants at different time point as mentioned before.

line 126: “none had negative results for spike IgG (< 50 AU/mL)” what does a negative result represent when there is no clear cut-off for the “protective” value?

Please consider renaming Figure titles across the manuscript in order to make it more straightforward and precise, i.e. sIgG1, sIgG2 are a bit misleading in particular because the Spike protein of SARS-CoV-2 consists of S1 and S2 subunits.

line 141: the outliers previously tested positive should not be considered at all if this study explores the immunogenicity of vaccines.

The present Table1 should be restyled by presenting IgG values at 6 and 8 months one next another in the same line to make it more clear and allow easier comparison, instead of having it separately one under the other.

In general, not necessary to divide each analysis by subsection title, a tab at the beginning of the paragraph would be enough; it would become friendlier environment for the reader to comprehend the presented information.

What are Fig 3 and Fig4 presenting - the regression slopes or correlation? there seem to be discrepancy between Figure titles and text (lines 190-191 and line 195)?

Please consider changing names of Figures with less wording and to be more concise, i.e. “Figure 3. A linear correlation between spike IgG level at 3 months and at 6 months (panel A), at 8 months (panel B), and at 6 months and at 8 months (panel C), stratified by age group” or similar.

Figures: please make correct labeling of the values of the variables presented in the Figures (female and male, and not using the study codes: 0 and 1)

line 211: regarding the predictive accuracy - this part is a bit challenging, using data that was used to derive formula for prediction, as a comparison and for the evaluation of its accuracy doesn’t sound quite solid. It seems quite reasonable that in this situation and on the identical population, the prediction would fit as good as it does here but this doesn’t allow for prediction in other population? This is a particular issue since there are several publications using more sophisticated statistical techniques and are still arguing this. Please consult the following reading doi.org/10.1371/journal.pone.0183250 and perhaps reconsider presenting this analysis.

Discussion: needs to be presented in more precise and clear way. Also, better comparison should be made with the previous findings.

line 233: doesn’t seem to be the main finding of this study , it’s actually a side-finding, I feel it should not be listed here.

line 268: it sounds too strong this statement, in particular because there are different recommendations and guidelines about adjusting this period for introducing the third dose, based on the specific characteristics of the population (particularly fragile, elderly, multi-comorbid, etc.) please consider rephrasing this.

It is highly unusual to have tables (3 and 4) with the results (?) presented in the Discussion section. Please describe these on the appropriate place and just refer to the results when discussing it in the Discussion section.

Conclusions: should be reinforced and clear message should be presented.

Conflict of interest: needs statement whether the company that attributed grant to the author was involved in any way and in the study design in particular since their machine was used.

Acknowledgment to the study participants would be nice.

6. PLOS authors have the option to publish the peer review history of their article (what does this mean?). If published, this will include your full peer review and any attached files.

Reviewer #1: No

Reviewer #2: No

Reviewer #3: No

---

## [Author Response · Author response to Decision Letter 0]

28 Apr 2022

We thank the reviewers for their thoughtful and constructive comments. In the following itemized response, reviewer comments are in bold italic black. Our responses are in blue, and changes to the revised manuscript are in red.

To the Reviewer #1: 

The manuscript by Takeuchi and colleagues shows statistics-supported prediction of SARS-CoV-2 Spike IgG levels 6 and 8 months after two doses of a COVID-19 mRNA vaccine in a healthy population. Overall the paper is written well and provides a useful and easy method to predict individual evolution of antibody titers, with important clinical implications, in particular the possibility of individual scheduling of a third vaccine dose.

RESPONSE: Thank you for the positive comments.

Major comments:

1. It’s not clear whether the enrolled participants were all naïve at the time of the vaccination or some of them had a SARS-CoV-2 infection before the first dose. This needs to be better explained as the two groups may have different starting titers and different decay kinetics. Does this also explain the great variability of sIgG2 titers shown in Figure 1? If numbers support a further sub-analysis, it would be interesting to assess the prediction in the two groups, separately.

RESPONSE: Thank you for these important comments. We also received the same comments from Reviewer #3. Accordingly, for this revised manuscript, we performed an additional analysis of nucleocapsid IgG, based on nucleocapsid IgG levels at 3, 6, and 8 months in 251, 250, and 249 subjects, respectively (One participant had only one examination at 3 months, and another had examinations at 3 and 6 months.). We found two subjects who had nucleocapsid IgG levels > 1.40 S/C, which was suggestive of the previous COVID-19 infections. We excluded these subjects from the linear regression analysis.

Lines 89-97:

To exclude the possibility of previous SARS-CoV-2 infection, we also measured IgG antibodies against the nucleocapsid protein of SARS-CoV-2 (Abbott Diagnostics) at 3, 6, and 8 months in all subjects. IgG against nucleocapsid protein (N) of SARS-CoV-2 in sera was measured by an ARCHITECT SARS-CoV-2 IgG assay on Architect system (Abbott Laboratories). The presence or absence of nucleocapsid IgG antibodies to SARS-CoV-2 was determined by comparing the chemiluminescent relative light unit in the reaction to the calibrator relative light unit, which is calculated as an index (S/C). According to the instruction of IgG (N) assay, the cut-off index is 1.40 S/C (https://www.fda.gov/media/137383/download).

Lines 140-141:

Nucleocapsid IgG of > 1.40 S/C, which was suggestive previous COVID-19 infection, was observed two subjects.

Lines 165-169:

Of two outliers, one had symptomatic SARS-CoV-2 infection with positive results from a reverse transcriptase-polymerase chain reaction (PCR) at 5 months and nucleocapsid IgG of 3.47 S/C at 6 months after the second BNT162b2 injection. The other denied any symptoms related to SARS-CoV-2 infection, and nucleocapsid IgG was < 1.40 S/C at all three time points.

Lines 174-178:

If we excluded the aforementioned two outliers and another subject whose nucleocapsid IgG at 3 months of 6.43 S/C, correlation coefficients further improved to 0.91 between 3 and 6 months and 0.86 between 3 and 8 months. Since two had a definite breakthrough infection and the other was suspected of a potential infection, we excluded these three cases from linear regression analysis.

2. Authors should better describe the rationale for performing the age sub-analysis defining 50 years as the cut-off for the comparison. Being the median age of 39, I would expect that there are not many participants aged greater than 50. How many healthcare workers belonged to the two groups? Would the results change if you define another age cut-off, such as the median age or 40 years?

RESPONSE: Thank you for the thoughtful comments. We just used 50 years as the cut-off, as in the previous study (PLoS ONE 2021;16:e0257668). We agreed that it did not make sense, and this time, we used new age cut-off, the median age in this revised manuscript.

Lines 103-104:

Subjects were divided into two groups according to median values of age (below 39 years or above 39 years) or gender.

Lines 200-202:

Linear regression analysis was performed separately for age groups stratified by the median age (39 years) (Fig.3) and gender (Fig. 4). There were no significant differences in regression slope and intercept between age ≥ 39 years and age < 39 years in any comparisons.

3. Using a cut-off of 300 resulted in a very low specificity at 6 and 8 months, which may raise concern on the usefulness of performing such a test at these later timepoints. This should be commented in the discussion. Can you perform a ROC analysis to define a new cut-off, which increases specificity and does not lower sensitivity too much? How would this change your prediction nomograms?

RESPONSE: Thank you for these comments and suggestions. We performed an ROC analysis and made a new table showing diagnostic accuracy of each cut-off value (Table 3). In the discussion, we also discussed a potential reason why specificity was low.

Line 122:

We also performed ROC analysis to determine the best cut-off values.

Lines 215-223:

Table 3 shows diagnostic accuracies of several cut-off values from equations for discriminating spike IgG of ≥ 300 AU/mL or < 300 AU/mL at 6 months and 8 months. Predicted spike IgG of ≥ 300 AU/mL or < 300 AU/mL at 6 months had 98% sensitivity, 47% specificity, and 94% accuracy for discriminating subjects whose observed spike IgG levels at 6 months were ≥ 300 AU/mL or < 300 AU/mL. Corresponding values of predicted spike IgG at 8 months were 97%, 70%, and 93%, respectively. ROC analysis revealed that the best cut-off value of predicted spike IgG levels at 6 months was 474 AU/mL with 89% sensitivity and 100% specificity. The corresponding best cut-off value of spike IgG at 8 month was 390 AU/mL with 87% sensitivity and 91% specificity.

Lines 324-331:

Prediction of spike IgG of ≥ 300 AU/mL or < 300 AU/mL at 6 months using the formula (Predicted log spike IgG at 6 months = 0.92 X log spike IgG at 3 months – 0.23) had a higher sensitivity (98%) but lower specificity (47%) for discriminating observed spike IgG of ≥ 300 AU/mL or < 300 AU/mL at 6 months. If we used higher cut-off values (450 AU/mL), specificity (96%) increased with some loss of sensitivity (90%). The reason was because predictions were usually noisy. Quite often when observed values were relatively low, they were lower than predicted values, but when observed values were relatively high, they were higher than predicted values [23].These trends are clearly shown in S1 Fig.

4. Another type of nomogram could be more useful to show at which month after vaccination you expect to reach the cut-off level (here 300) based on titers measured at 3 months. This would better and more directly support an individual scheduling for a third dose.

RESPONSE: Thank you for the thoughtful suggestions. Accordingly, we made a new nomogram (Table 5). 

Lines 255-258:

Table 5 is another nomogram showing the month at which we expect to reach the cut-off level of spike IgG (300 AU/mL) after two-dose vaccination. For example, if spike IgG at 3 months was 800 AU/mL, expected elapsed time to reach spike IgG level < 300 AU/mL would be 5.5 months.

Lines 347-349:

Another nomogram (Table 5) shows at which month after vaccination one expects to reach the pre-defined spike IgG cut-off level (300 AU/mL) based on levels at 3 months. This nomogram supports individual scheduling for a third dose.

5. Many studies have shown the decay of Spike-IgG titers after vaccination. Although beyond the scope of this study, a comparison of the equations obtained applying the same method described in this study to other published cohorts will be useful to further support the accuracy of the predictions for healthy individuals of different ethnicities.

RESPONSE: We could not find any datasets that allowed us to apply our regression model to determine whether it is correct. Thus, this point was discussed in the study limitations.

Lines 357-360:

Third, although we generated simple equations, a comparison of the equations obtained applying the same method described in this study to other cohorts will be useful to further support the accuracy of the predictions for healthy individuals of different ethnicities.

Minor issues:

1. Check spelling of “Spike” throughout the text as it should be always capitalized.

RESPONSE: We corrected it.

2. The short terms sIgG1, sIgG2 and sIgG3 remind “serum IgG isotypes 1, 2 and 3”. I would change them into either TP1, T2, TP3… (timepoint) or even with the actual timing (3 mths, 6 mths, 8 mths). There’s no need to specify Spike IgG as it is clear from the y axis.

RESPONSE: We used TP0, TP0.5, TP3, TP6, and TP8 in Fig 1.

3. Please provide a Source Data file for each figure and table or a unique supplemental summary table with IgG titers for each participants at different time points and information about the age and gender category used for the sub-analyses.

RESPONSE: We uploaded the latest data file. 

Reviewer #2: The study conducted by Takeuchi et al., entitled " Prediction of Spike IgG Levels in Japanese Workers 6 and 8 months after the Second Dose of BNT162b2 Vaccine" was designed to detect the levels of anti-S IgG at 3, 6, and 8 months after two doses of BNT162b2 Vaccine. The authors hypothesized that levels of anti-S IgG at three months after two doses predict the levels after 6 and 8 months based on which one can determine if a third dose is needed within a fixed time frame.

This study is an extension of the previous study by the authors where they hypothesized that the reactogenicity after two doses of the BNT162b2 vaccine may predict spike IgG antibody levels.

Results of this study are consistent with current literature in the field. However, on the whole, the paper is not suitable for publication in its present form.

RESPONSE: Thank you for your time and effort to review this manuscript.

Specific Comments:

At the time of the study, booster (3rd dose) was already approved and in use in many countries. It is now common globally. Hence studying IgG levels at 8 months after the second dose make the significance of this study questionable. Also, it is important to perform measurements of neutralizing antibody titers, which are a better reflection of protective immune responses. Measuring total S-IgG levels after 3 months do not offer a complete picture of the virus neutralization capacity. And, total S-IgG levels cannot be the sole predictor for adequate immunity post-vaccination and is not sufficient to determine who needs a 3rd vaccine dose.

RESPONSE: Thank you for this important criticism. Although several governments decided to inject a third vaccine dose sooner than 8 months after the second dose, spike IgG data at 8 months after the second dose are important in order to monitor decay kinetics of spike IgG in healthy subjects. We totally agree that neutralizing antibody levels are a better reflection of immune response than spike IgG levels. However, neutralizing antibody assays are not easy and are costly. These points are described in the study limitations.

Lines 364-370:

Finally and more importantly, although spike IgG levels correlate with vaccine efficacy [30], we did not measure neutralizing antibody titers, which are better reflection of protective immune responses after vaccination. However, neutralizing antibody assays are costly and not fully automatic, resulting in low penetration in the general clinical field. They also require additional equipment. We believe that spike IgG monitoring could offer a relatively low-cost monitoring strategy in all situations [22].

The authors need to be consistent using the terms SARS-CoV-2 and COVID-19.

The authors use the terms “injections, vaccination, and dose” and “spike IgG titers, values and levels” interchangeably. There needs to be a consistency in the use of specific terms.

RESPONSE: Thank you for these important comments. We revised the manuscript insofar as possible.

Abstract

Page2, Line 15: The term full vaccination needs to be properly defined, especially since the authors did not assess the antibody levels after the 3rd vaccine dose. There is not a clear conclusion about the results shown in the manuscript

RESPONSE: We corrected them as suggested.

Line 16:

that spike IgG levels at 3 months after two doses of the BNT162b2 vaccine

Lines 33-35:

We conclude that predictive formulae using spike IgG levels at 3 months after two-dose vaccination with BNT162b2 reliably estimate subsequent spike IgG levels up to 8 months and provide useful information in terms of vaccination booster timing.

Introduction

Page 4, lines 51-53: The authors make the following statement “Since we found that spike IgG titers two-weeks after the second dose of BNT162b2 vaccine vary widely, we hypothesized that not all subjects should need a third vaccine dose within a fixed time window”. The authors likely need more evidence to substantiate this hypothesis.

RESPONSE: We modified the sentences.

Lines 52-55:

In our previous study, spike IgG levels two-weeks after the second dose of BNT162b2 ranged from 2,826 to 70,272 AU/mL in 67 COVID-19 naïve healthy Japanese healthcare workers [21], meaning that not all subjects may need a third vaccine dose within a fixed time window. 

Materials and Methods

Antibody test

Authors need to describe the Abbott Antibody test in detail. How are the assay cut-off values or S and N-IgG calculated? 

RESPONSE: We revised these sentences.

Lines 85-97:

The titer of IgG against S protein’s receptor biding domain (RBD) in sera from blood sample was measured by an ARCHITECT SARS-CoV-2 IgG II Quant assay on Architect system (Abbott Laboratories, Abbott Park, IL, USA). According to the instruction of IgG (RBD) assay, the cut-off index is 50.0 arbitrary units/mL (AU/mL) (https://www.fda.gov/media/146371/download). To exclude the possibility of previous SARS-CoV-2 infection, we also measured IgG antibodies against the nucleocapsid protein of SARS-CoV-2 (Abbott Diagnostics) at 3, 6, and 8 months in all subjects. IgG against nucleocapsid protein (N) of SARS-CoV-2 in sera was measured by an ARCHITECT SARS-CoV-2 IgG assay on Architect system (Abbott Laboratories). The presence or absence of nucleocapsid IgG antibodies to SARS-CoV-2 was determined by comparing the chemiluminescent relative light unit in the reaction to the calibrator relative light unit, which is calculated as an index (S/C). According to the instruction of IgG (N) assay, the cut-off index is 1.40 S/C (https://www.fda.gov/media/137383/download).

Page 6, line 84: clarify the negative cut-off value

RESPONSE: We eliminated the negative cut-off value.

Statistical analysis

Table 1: A Student’s t test is not the appropriate test for this data set. The statistics should have been calculated using an ANOVA or non-parametric test with corrections for multiple comparisons.

RESPONSE: This is not t-test but t-value, and it was used to obtain a p-value for each covariate in the linear regression analysis.

Page 6, lines 101-103 The authors need to add a proper reference to the study they used to determine the S-IgG cut-off value of 300 AU/ml to stratify the study subjects.

RESPONSE: Thank you. We added another manuscript to support the assertion that a spike IgG level of 300 AU/mL is an important cut-off value.

Lines 114-119:

Although there were no definite cut-off values of spike IgG levels for adequate immunity, a large Israeli study proposed that subjects whose spike IgG levels were ≤ 300 AU/mL were more likely to become re-infected with COVID-19, than those whose antibody levels were > 300 AU/mL [10]. A position paper also addressed that spike IgG levels < 280 – 300 AU/mL for the Abbott IgG II assay were defined as non-responder patients with cancer [22]. Thus, we used a spike IgG of 300 AU/mL as a cut-off value. 

Results

All Figures need better labeling throughout and added to the manuscript in a printable resolution.

RESPONSE: Thank you. We corrected this as suggested.

Page 7, Line 112: Inclusion criteria needs to be described the methods

RESPONSE: Thank you. We specified the inclusion criteria.

Lines 70-72:

Inclusion criteria were subjects who had received the first dose of BNT162b2 mRNA Covid-19 vaccine from March 22 to March 26, 2021 and the second dose from April 12 to April 23, 2021. 

Line 114, the authors need to describe the details of their previous study in the methods and not in the results section

RESPONSE: Thank you. 

Figure 1

Figure 1 labeling is confusing. sIgG 1, 2, and 3 usually refers to the subtypes and hence the X-axis labels need to be modified properly to reflect the time points. S-IgG AU/ml data should be log-transformed or presented on a log scale for low-end readability.

RESPONSE: Thank you. We corrected Fig 1 as suggested.

Figures 3 and 4

Correlation coefficients need to be included 

RESPONSE: Thank you, but this was a linear regression; thus, we provided the slope and intercept.

Discussion

Page 12, line 236: define the terms “excellent and good accuracy”.

RESPONSE: We deleted excellent and good, and we included actual numbers.

Lines 270-273:

(3) Predicted spike IgG levels of 300 AU/mL estimated from spike IgG levels at 3 months accurately stratified subjects whose observed spike IgG levels were ≥ 300 AU/mL or < 300 AU/mL at 6 months with 94% accuracy and at 8 months with 93% accuracy.

Page 13, lines 255-260: the authors need to explain how the current study compares to the study from Israel. 

RESPONSE: We just want to derive some cut-off values for the prediction.

Lines 291-300:

Although there is no definite cut-off value of spike IgG to prevent COVID-19 infection, Kertes et al. [10] determined the relationship between spike IgG levels and subsequent PCR-confirmed infection in a large number of Israeli subjects. The authors reported that the proportion of participants with positive PCR results was 1.2% for those with spike IgG below 150 AU/mL, 1.3% with spike IgG levels between150 AU/mL and 300 AU/mL, and 0.2% of those with spike IgG levels above 300 AU/mL. A recent position paper about cancer patients recommends a third vaccination in non-responder patients, defined as those whose spike IgG levels < 40 binding antibody units (BAU) / mL (< 280 – 300 AU/mL for the Abbott IgG II assay). Thus, we used a spike IgG cut-off value of 300 AU/mL as a reflection of a high risk of breakthrough infection and non-responder after the vaccination.

Page 13, line 263: This sentence is confusing and needs to be re-written

RESPONSE: We rephrased it.

Lines 303-304:

Overall, median values of spike IgG 6 months after vaccination became one-third of levels at 3 months, findings consistent with previous studies [4, 7-10].

Page 14, lines 267-269: There is no sufficient evidence in this study to substantiate this claim

RESPONSE: Thank you. We deleted this sentence as suggested.

Page 14, line 279: line 286: explain “decay”

RESPONSE: We rephrased this.

Lines 319-321:

These results suggest that after reaching peak values, spike IgG levels decrease at a relatively constant rate, irrespective of age or gender.

The sentence, including another “decay”, was omitted in this revised manuscript.

Page 15, lines 293-294 term “after vaccination” needs to be defined.

RESPONSE: Thank you. We rephrased it.

Lines 334-336:

we think that best time to predict future spike IgG levels may be from 1 to 3 months after the second vaccination.

Page 15, lines 313-315: The authors need to explain how they came with the S-IgG cut-ff value of 300AU/ml to stratify the study subjects.

RESPONSE: The answer is the same as in the previous query.

Lines 291-300:

Although there is no definite cut-off value of spike IgG to prevent COVID-19 infection, Kertes et al. [10] determined the relationship between spike IgG levels and subsequent PCR-confirmed infection in a large number of Israeli subjects. The authors reported that the proportion of participants with positive PCR results was 1.2% for those with spike IgG below 150 AU/mL, 1.3% with spike IgG levels between150 AU/mL and 300 AU/mL, and 0.2% of those with spike IgG levels above 300 AU/mL. A recent position paper about cancer patients recommends a third vaccination in non-responder patients, defined as those whose spike IgG levels < 40 binding antibody units (BAU) / mL (< 280 – 300 AU/mL for the Abbott IgG II assay). Thus, we used a spike IgG cut-off value of 300 AU/mL as a reflection of a high risk of breakthrough infection and non-responder after the vaccination.

 

Reviewer #3: A study by Takeuchi and colleagues explored temporal changes in the spike IgG titers after two doses of BNT162b2 vaccine and tried to determine whether spike IgG levels at 3 months can predict levels 6 or 8 months after full vaccination.

In general, this topic is highly relevant and the recruitment of study participants is done correctly. Surely, the high value of this study is the longitudinal collection of data during 3 (5) points in time, but the presentation of the results is pretty modest, with several critical segments that need to be improved before considering this paper for final publication. Several parts of the manuscript need to be rephrased.

RESPONSE: We appreciate your time and effort to review our manuscript.

I find the title of the paper a bit inappropriate since IgG levels are actuality measured at 6 and 8 months and not predicted. What authors did is that they derived formula for predicting values (using measured vales as dependent variables in the linear regression) so this is not main finding of the study and it might be misleading, please consider rephrasing it.

RESPONSE: Thank you. We rephrased the title as follows:

“Temporal changes in spike IgG levels after two doses of BNT162b2 vaccine in Japanese healthcare workers: Do spike IgG levels at 3 months predict levels 6 or 8 months after the vaccination?”

Point-by-point comments are listed below.

Introduction: this part needs further elaboration on the background of this topic, stronger rationale why this kind of study was needed.

RESPONSE: Thank you for these important comments. We revised Introduction as suggested.

line 46: “Although new variants of concern that escape the immune system are another potential cause of vaccine ineffectiveness…” please replace ineffectiveness with “lower effectiveness” since it’s incorrect.

RESPONSE: Thank you. We corrected it.

Lines 47-48:

lower vaccine effectiveness

Methods: why authors opted for measurement at 3 months? What is the rationale behind this decision?

RESPONSE: We described the rationale in the Introduction.

Lines 55-59:

A recent position paper regarding cancer patients recommends measuring spike IgG at 3 months in responder patients (spike IgG 3 to 4 weeks after second dose vaccination > 1,800 AU/mL) [22]. We hypothesized that spike IgG level at 3 month is a good time to assess the kinetics of waning immunity because all vaccinees have already reached peak levels of spike IgG. 

Why there was a cut-off at 50 AU/mL? further, a reference would be helpful.

RESPONSE: We added references.

Lines 88-89:

the cut-off index is 50.0 arbitrary units/mL (AU/mL) (https://www.fda.gov/media/146371/download).

line 83: the authors state that IgG anti-N was measured “at least one time”, this is a potential issue and remains unclear since it may compromise the level of IgG measured? these should have been measured at all time points when IgG anti-S were measured? Without this how is that the authors may be confident that these IgG anti-S are the effect of the vaccine and not combined effect with (a)symptomatic COVID?

RESPONSE: Thank you for the thoughtful comments. Accordingly, for this revised manuscript, we performed an additional analysis of nucleocapsid IgG, based on nucleocapsid IgG levels at 3, 6, and 8 months in 251, 250, and 249 subjects, respectively (One participant had only one examination at 3 months, and another had examinations at 3 and 6 months.). We found two subjects who had nucleocapsid IgG levels > 1.40 S/C, which was suggestive of the previous COVID-19 infections. We excluded these two subjects and another who was outlier in Fig 2. from the linear regression analysis.

Lines 89-97:

To exclude the possibility of previous SARS-CoV-2 infection, we also measured IgG antibodies against the nucleocapsid protein of SARS-CoV-2 (Abbott Diagnostics) at 3, 6, and 8 months in all subjects. IgG against nucleocapsid protein (N) of SARS-CoV-2 in sera was measured by an ARCHITECT SARS-CoV-2 IgG assay on Architect system (Abbott Laboratories). The presence or absence of nucleocapsid IgG antibodies to SARS-CoV-2 was determined by comparing the chemiluminescent relative light unit in the reaction to the calibrator relative light unit, which is calculated as an index (S/C). According to the instruction of IgG (N) assay, the cut-off index is 1.40 S/C (https://www.fda.gov/media/137383/download).

Lines 140-141:

Nucleocapsid IgG of > 1.40 S/C, which was suggestive previous COVID-19 infection, was observed two subjects.

Lines 165-169:

Of two outliers, one had symptomatic SARS-CoV-2 infection with positive results from a reverse transcriptase-polymerase chain reaction (PCR) at 5 months and nucleocapsid IgG of 3.47 S/C at 6 months after the second BNT162b2 injection. The other denied any symptoms related to SARS-CoV-2 infection, and nucleocapsid IgG was < 1.40 S/C at all three time points.

Lines 174-178:

If we excluded the aforementioned two outliers and another subject whose nucleocapsid IgG at 3 months of 6.43 S/C, correlation coefficients further improved to 0.91 between 3 and 6 months and 0.86 between 3 and 8 months. Since two had a definite breakthrough infection and the other was suspected of a potential infection, we excluded these three cases from linear regression analysis.

Statistical analyses: why was the data log-transformed? Skewed distribution? Any tests and visualization techniques used for the exploration of data normality (Q-Q plot, K-S test, Shapiro-Wilk test, etc.)?

RESPONSE: Thank you. We incorporated this.

Lines 104-106:

For the main analysis, spike IgG levels were log-transformed due to skewness of the data distribution which was verified using the Shapiro-Wilk test and Q-Q plots.

line 91: the authors state that “Subjects were divided into two groups according to age (below 50 years or 50 years and above)”. What is the rationale behind this? Which hypothesis is tested there? This is much unclear, especially because a table with general description of the participants is missing.

RESPONSE: Thank you for these thoughtful comments. We just used 50 years as the cut-off as in previous study (PLoS ONE 2021;16:e0257668). We agree that it did not make sense, so we used a new age cut-off here, the median age in this revised manuscript.

Lines 103-104

Subjects were divided into two groups according to median values of age (below 39 years or above 39 years) or gender.

Lines 200-202

Linear regression analysis was performed separately for age groups stratified by the median age (39 years) (Fig.3) and gender (Fig. 4). There were no significant differences in regression slope and intercept between age ≥ 39 years and age < 39 years in any comparisons.

Results: this section needs major improvements, it need to be rephrased and presented in more concise, straightforward and clear way.

A Table with the description of general characteristics of the study participants is highly needed here. Also, there no mention of the potential comorbidities, medication used, vaccine reactogenicity, which all may have affected the IgG levels measured here?

RESPONSE: We made Table 1 showing general characteristics of the study participants. We also added the following description.

Lines 138-139:

None had immunocompromised, and none had taken immunosuppressive medications.

On which department the study participants work on, are they involved in the treatment of the COVID patients? This might have potentially exposed them to the virus and booster their IgG levels, especially problematic if IgG anti-N were not measured for all participants at different time point as mentioned before.

RESPONSE: Thank you for these important comments. Approximately 30% of subjects were working in COVID-19 wards. We included this information in the manuscript as well as in Table1.

Lines 139-140:

73 HCWs (29%) were working in COVID-19 wards.

line 126: “none had negative results for spike IgG (< 50 AU/mL)” what does a negative result represent when there is no clear cut-off for the “protective” value?

RESPONSE: We rephrased the sentence.

Line 153:

However, none had a spike IgG < 50 AU/mL even at 8 months.

We also provided a reference for 50 AU/mL.

Lines 88-89:

the cut-off index is 50.0 arbitrary units/mL (AU/mL) (https://www.fda.gov/media/146371/download).

Please consider renaming Figure titles across the manuscript in order to make it more straightforward and precise, i.e. sIgG1, sIgG2 are a bit misleading in particular because the Spike protein of SARS-CoV-2 consists of S1 and S2 subunits.

RESPONSE: Thank you for these important suggestions. We revised the manuscript accordingly.

line 141: the outliers previously tested positive should not be considered at all if this study explores the immunogenicity of vaccines.

RESPONSE: We omitted three cases for possible COVID-19 infection according to nucleocapsid IgG results and correlation analysis.

Lines 174-178:

If we excluded the aforementioned two outliers and another subject whose nucleocapsid IgG at 3 months of 6.43 S/C, correlation coefficients further improved to 0.91 between 3 and 6 months and 0.86 between 3 and 8 months. Since two had a definite breakthrough infection and the other was suspected of a potential infection, we excluded these three cases from linear regression analysis.

The present Table1 should be restyled by presenting IgG values at 6 and 8 months one next another in the same line to make it more clear and allow easier comparison, instead of having it separately one under the other.

RESPONSE: Done.

In general, not necessary to divide each analysis by subsection title, a tab at the beginning of the paragraph would be enough; it would become friendlier environment for the reader to comprehend the presented information.

RESPONSE: We revised it.

What are Fig 3 and Fig4 presenting - the regression slopes or correlation? there seem to be discrepancy between Figure titles and text (lines 190-191 and line 195)?

RESPONSE: Thank you for the thoughtful comments. This was a linear regression analysis because our main purpose was to see whether there was an interaction between log transformed spike IgG levels and age (or gender).

Please consider changing names of Figures with less wording and to be more concise, i.e. “Figure 3. A linear correlation between spike IgG level at 3 months and at 6 months (panel A), at 8 months (panel B), and at 6 months and at 8 months (panel C), stratified by age group” or similar.

RESPONSE: Thank you. We revised this as suggested.

Lines 206-210:

Fig 3. A linear regression analysis between spike IgG levels at 3 and 6 months (panel A), at 8 months (panel B), and at 6 and 8 months (panel C), stratified by age group

Fig 4. A linear regression analysis between spike IgG level at 3 and 6 months (panel A), at 8 months (panel B), and at 6 and 8 months (panel C), stratified by gender

Figures: please make correct labeling of the values of the variables presented in the Figures (female and male, and not using the study codes: 0 and 1)

RESPONSE: Thank you. We revised this as suggested.

line 211: regarding the predictive accuracy - this part is a bit challenging, using data that was used to derive formula for prediction, as a comparison and for the evaluation of its accuracy doesn’t sound quite solid. It seems quite reasonable that in this situation and on the identical population, the prediction would fit as good as it does here but this doesn’t allow for prediction in other population? This is a particular issue since there are several publications using more sophisticated statistical techniques and are still arguing this. Please consult the following reading doi.org/10.1371/journal.pone.0183250 and perhaps reconsider presenting this analysis.

RESPONSE: Thank you for the thoughtful comments. We read the paper that you mentioned, and found that the content was not easy to understand. Anyway, we incorporated VEcv. We think that Lin’s concordance correlation coefficient (CCC) is another metric to quantify accuracy; thus, we also included Lin’s CCC in this revised manuscript.

Lines 122-125:

To evaluate the accuracy of predicted spike IgG at 6 and 8 months, we calculated variance explained by predictive models based on cross-validation (VEcv) [23] and Lin’s concordance correlation coefficient [24].

Lines 232-237:

S1 Fig. shows correlation between observed spike IgG levels and predicted spike IgG levels at 6 months and 8 months. VEcv and Lin’s CCC were 83.2% and 0.91 at 6 months. The corresponding values were 73.7% and 0.85 at 8 months after two-dose vaccination.

S1 Fig. A linear correlation between observed spike IgG level (x-axis) and predicted spike IgG level (y-axis) at 6 months (panel A), at 8 months (panel B)

Discussion: needs to be presented in more precise and clear way. Also, better comparison should be made with the previous findings.

RESPONSE: Thank you. We revised this to the extent possible.

line 233: doesn’t seem to be the main finding of this study , it’s actually a side-finding, I feel it should not be listed here.

RESPONSE: Thank you. We deleted the sentence.

line 268: it sounds too strong this statement, in particular because there are different recommendations and guidelines about adjusting this period for introducing the third dose, based on the specific characteristics of the population (particularly fragile, elderly, multi-comorbid, etc.) please consider rephrasing this.

RESPONSE: Thank you. We agree with your comments and omitted the sentence. We rephrased the following text.

Lines 307-311:

Although there are different recommendations and guidelines about adjusting the period for introducing the third dose, based on specific characteristics of the population, especially in fragile, elderly, and multi-comorbid populations, an individualized vaccine schedule should be established to maintain effective prevention of breakthrough infection under limited vaccine supply.

It is highly unusual to have tables (3 and 4) with the results (?) presented in the Discussion section. Please describe these on the appropriate place and just refer to the results when discussing it in the Discussion section.

RESPONSE: Thank you. We deleted the previous Table 3 in this revised manuscript. We made another nomogram (Table 5) according to a suggestion from Reviewer #1. We replaced the tables in the Results suction and discussed them in the clinical implications.

Lines 239-240:

Using equations, we made a nomogram showing minimum spike IgG levels at 3 months in order to maintain certain levels of spike IgG at 6 and 8 months (Table 4).

Lines 255-258:

Table 5 is another nomogram showing the month at which we expect to reach the cut-off level of spike IgG (300 AU/mL) after two-dose vaccination. For example, if spike IgG at 3 months was 800 AU/mL, expected elapsed time to reach spike IgG level < 300 AU/mL would be 5.5 months.

Lines 347-349:

Another nomogram (Table 5) shows at which month after vaccination one expects to reach the pre-defined spike IgG cut-off level (300 AU/mL) based on levels at 3 months. This nomogram supports individual scheduling for a third dose.

Conclusions: should be reinforced and clear message should be presented.

RESPONSE: Thank you. We rephrased the Conclusion as follows.

Lines 373-376:

Spike IgG levels 6 and 8 months after two-dose vaccination with BNT162b2 are closely correlated with spike IgG levels at 3 months. Thus, spike IgG levels at 3 months predict subsequent spike IgG levels with high accuracy. Nomograms generated from results of this study provide useful information for individual scheduling for third vaccine dose.

Conflict of interest: needs statement whether the company that attributed grant to the author was involved in any way and in the study design in particular since their machine was used.

Acknowledgment to the study participants would be nice.

RESPONSE: Thank you. We incorporated.

Lines 378-383:

Competing interests:

MT received a research grant from Abbott. Other authors have no conflicts of interest. This does not alter our adherence to PLOS ONE policies on sharing data and materials.

Acknowledgment:

We thank all study participants.

---

## [Decision Letter · Decision Letter 1]

31 May 2022

PONE-D-22-01729R1Temporal changes in spike IgG levels after two doses of BNT162b2 vaccine in Japanese healthcare workers: Do spike IgG levels at 3 months predict levels 6 or 8 months after vaccination?PLOS ONE

Dear Dr. Takeuchi,

Thank you for submitting your manuscript to PLOS ONE. After careful consideration, we feel that it has merit but does not fully meet PLOS ONE’s publication criteria as it currently stands. Therefore, we invite you to submit a revised version of the manuscript that addresses the points raised during the review process. The comments seem minor. Please perform one more round of the review process.

We look forward to receiving your revised manuscript.

Kind regards,

Etsuro Ito

Academic Editor

PLOS ONE

Journal Requirements:

Reviewers' comments:

Reviewer's Responses to Questions

**Comments to the Author**

1. If the authors have adequately addressed your comments raised in a previous round of review and you feel that this manuscript is now acceptable for publication, you may indicate that here to bypass the “Comments to the Author” section, enter your conflict of interest statement in the “Confidential to Editor” section, and submit your "Accept" recommendation.

Reviewer #1: All comments have been addressed

Reviewer #3: (No Response)

2. Is the manuscript technically sound, and do the data support the conclusions?

Reviewer #1: Yes

Reviewer #3: Partly

3. Has the statistical analysis been performed appropriately and rigorously? 

Reviewer #1: Yes

Reviewer #3: Yes

4. Have the authors made all data underlying the findings in their manuscript fully available?

Reviewer #1: Yes

Reviewer #3: Yes

5. Is the manuscript presented in an intelligible fashion and written in standard English?

Reviewer #1: Yes

Reviewer #3: Yes

6. Review Comments to the Author

Reviewer #1: (No Response)

Reviewer #3: Dear Authors,

Thank you for addressing comments listed in the first round of the review process. I find this new modified version to be greatly improved even though I feel that there are still some minor issues left. Please check my comments and try to provide additional modifications where appropriate or discuss the rationale for opposing.

Introduction:

lines 51-52: “Due to limited vaccine supply, an individualized booster vaccine schedule is mandatory to conserve vaccine doses and to prioritize them to high-risk patients or developing countries.” try replacing “mandatory” with synonyms (i.e., advisably/needed/recommended), since mandatory seems too strong, or please provide explanation if inappropriate.

in lines 55-57 the Authors explained that the rationale for choosing the 3rd month for measuring the Ab level based on a study [ref 22] that are actually the recommendations applied to patients with cancer. I'm not sure whether this is appropriate since is a particular population there in which (due to the disease and treatment) the Ab dynamics might be different? Also, the following sentence: “We hypothesized that spike IgG level at 3 month is a good time to assess the kinetics of waning immunity because all vaccinees have already reached peak levels of spike IgG.” lacks appropriate reference? Based on what these hypotheses were derived? This needs to be explained.

Methods:

The authors provided little satisfactory response to my previous comment regarding the cut-off values: ”Why there was a cut-off at 50 AU/mL? further, a reference would be helpful. RESPONSE: We added references.”

Unfortunately, the provided reference (https://www.fda.gov/media/146371/download) also contains little information regarding the cutoff values with and additional remark: “Interpretation of Results -The cutoff is 50.0 AU/mL. As with all analyte determinations, the result should be used in conjunction with information available from clinical evaluation and other diagnostic procedures.” I guess this is still unclear in the literature and without a consensus thus I feel these results should be interpreted with caution.

Other previous comment: “the authors state that “Subjects were divided into two groups according to age (below 50 years or 50 years and above)”. What is the rationale behind this? Which hypothesis is tested there? This is much unclear, especially because a table with general description of the participants is missing.” The Authors now have made modifications and provided RESPONSE: “Thank you for these thoughtful comments. We just used 50 years as the cut-off as in previous study (PLoS ONE 2021;16:e0257668). We agree that it did not make sense, so we used a new age cut-off here, the median age in this revised manuscript.” Even though it has some sense stratifying data based on the data characteristics, I'm still not sure whether it is appropriate? it’s still unclear which hypothesis was tested here?

Finally, this is still my major concern from the previous round: “line 211: regarding the predictive accuracy - this part is a bit challenging, using data that was used to derive formula for prediction, as a comparison and for the evaluation of its accuracy doesn’t sound quite solid. It seems quite reasonable that in this situation and on the identical population, the prediction would fit as good as it does here but this doesn’t allow for prediction in other population? This is a particular issue since there are several publications using more sophisticated statistical techniques and are still arguing this. Please consult the following reading doi.org/10.1371/journal.pone.0183250 and perhaps reconsider presenting this analysis.” the Authors provided RESPONSE: “Thank you for the thoughtful comments. We read the paper that you mentioned, and found that the content was not easy to understand. Anyway, we incorporated VEcv. We think that Lin’s concordance correlation coefficient (CCC) is another metric to quantify accuracy; thus, we also included Lin’s CCC in this revised manuscript.” Additionally, I agree with the comment 5 from the Reviewer #1 that said “a comparison of the equations obtained applying the same method described in this study to other published cohorts will be useful to further support the accuracy of the predictions for healthy individuals of different ethnicities.” In fact, the S1 Fig shows good correlation between predicted and measured values, which is expected when using the same dataset, as I previously explained. I feel this is another limitation of the study. The problem is not comparison to other ethnicity as listed in the lines 358-360 but the methodology (data) used for accuracy evaluation.

Results:

Table 1: please provide appropriate labeling of the Variables i.e. Age (years), median (IQR); Male, n (%), etc. What represents the Part-time occupation? it’s unclear, it doesn’t seem a position on which they work? should be explained or modified. I don’t feel “Nucleocapsid IgG at 3/6/8 months” values should be listed here in Table 1 and are a bit misleading since these are not characteristics of the participants but something that was measured in this study and also only 3 participants were positive on this test so median of 0.02 doesn’t tell much since many had value of 0?

Correlation coefficients (r) should be accompanied by corresponding p-values (lines 164, 165, etc.)

As per propositions of PLOS One, tables should be placed right after being first mentioned in the text (Table2).

I find information from the nomograms presented in Table 4 (and 5) to be very useful. But, how was the formula used for the Predicted elapsed month = 37.5 * (1 – (2.71 / log-transformed spike IgG at 3 month)) derived? There is no explanation on this?

Discussion:

I would be more satisfied if the authors used some of the studies listed in lines 285-287 “Several studies have measured short- [28, 29] and long-term changes [4-14] in spike IgG and neutralizing antibody levels up to 9 months after two doses of BNT162b2, and all studies have revealed that antibody levels drop over time.” to make more vivid discussion, i.e., to compare their results with those obtained by previous studies (since there are many available nowadays). Also, the results seem quite valuable but in the discussion I miss the answer on the major question “what might be the utility of these findings” and stronger statement on why these results are valuable for practice?

line 339 seems a bit misleading, these are more Public health implication rather than Clinical implications? this section should answer the question: What would these findings add if being implemented in practice?

7. PLOS authors have the option to publish the peer review history of their article (what does this mean?). If published, this will include your full peer review and any attached files.

Reviewer #1: No

Reviewer #3: No

---

## [Author Response · Author response to Decision Letter 1]

5 Jun 2022

We thank the reviewer for his/her thoughtful and constructive comments. In the following itemized response, reviewer comments are in bold italic black. Our responses are in blue, and changes to the revised manuscript are in red.

To the Reviewer #3: 

Dear Authors,

Thank you for addressing comments listed in the first round of the review process. I find this new modified version to be greatly improved even though I feel that there are still some minor issues left. Please check my comments and try to provide additional modifications where appropriate or discuss the rationale for opposing.

Thank you for the positive comments.

Introduction:

lines 51-52: “Due to limited vaccine supply, an individualized booster vaccine schedule is mandatory to conserve vaccine doses and to prioritize them to high-risk patients or developing countries.” try replacing “mandatory” with synonyms (i.e., advisably/needed/recommended), since mandatory seems too strong, or please provide explanation if inappropriate.

We replaced “mandatory” with “recommended”.

Page 4, line 52:

an individualized booster vaccine schedule is recommended to conserve

in lines 55-57 the Authors explained that the rationale for choosing the 3rd month for measuring the Ab level based on a study [ref 22] that are actually the recommendations applied to patients with cancer. I'm not sure whether this is appropriate since is a particular population there in which (due to the disease and treatment) the Ab dynamics might be different? Also, the following sentence: “We hypothesized that spike IgG level at 3 month is a good time to assess the kinetics of waning immunity because all vaccinees have already reached peak levels of spike IgG.” lacks appropriate reference? Based on what these hypotheses were derived? This needs to be explained.

We incorporated appropriate references and modified the sentence as follows.

Page 4, Lines 59-61:

We hypothesized that spike IgG level at 3 months is a good time to assess the kinetics of waning immunity because all vaccinees have already reached peak levels of spike IgG [4, 13, 25-27], and some studies found a decline in antibody levels at 3 months [4, 13, 14, 26, 28].

Methods:

The authors provided little satisfactory response to my previous comment regarding the cut-off values: ”Why there was a cut-off at 50 AU/mL? further, a reference would be helpful. RESPONSE: We added references.”

Unfortunately, the provided reference (https://www.fda.gov/media/146371/download) also contains little information regarding the cutoff values with and additional remark: “Interpretation of Results -The cutoff is 50.0 AU/mL. As with all analyte determinations, the result should be used in conjunction with information available from clinical evaluation and other diagnostic procedures.” I guess this is still unclear in the literature and without a consensus thus I feel these results should be interpreted with caution.

Thank you. We incorporated one reference to verify the diagnostic accuracy of the cut-off value (50 AU/mL) in this revised manuscript.

Page 6, line 91:

units/mL (AU/mL) (https://www.fda.gov/media/146371/download) [29].

Page 26, lines 519-522

29. Narasimhan M, Mahimainathan L, Araj E, Clark AE, Markantonis J, Green A, et al. Clinical Evaluation of the Abbott Alinity SARS-CoV-2 Spike-Specific Quantitative IgG and IgM Assays among Infected, Recovered, and Vaccinated Groups. J Clin Microbiol. 2021;59(7):e00388-21. doi: 10.1128/jcm.00388-21 PMID - 33827901.

Other previous comment: “the authors state that “Subjects were divided into two groups according to age (below 50 years or 50 years and above)”. What is the rationale behind this? Which hypothesis is tested there? This is much unclear, especially because a table with general description of the participants is missing.” The Authors now have made modifications and provided RESPONSE: “Thank you for these thoughtful comments. We just used 50 years as the cut-off as in previous study (PLoS ONE 2021;16:e0257668). We agree that it did not make sense, so we used a new age cut-off here, the median age in this revised manuscript.” Even though it has some sense stratifying data based on the data characteristics, I'm still not sure whether it is appropriate? it’s still unclear which hypothesis was tested here?

Since age was not a significant covariate for multivariate linear regression analysis (Table 2), we deleted the sentences regarding subgroup analysis using age and in the previous figure 3.

Page 6, lines 105-106:

groups according to median values of age (below 39 years or above 39 years) or gender.

Page 12, lines 205-207

Linear regression analysis was performed separately for age groups stratified by the median age (39 years) (Fig.3) and gender (Fig.3). There were no significant differences in regression slope and intercept between age ≥ 39 years and age < 39 years in any comparisons.

Page 12, lines 211-212:

Fig 3. A linear regression analysis between spike IgG levels at 3 and 6 months (panel A), at 8 months (panel B), and at 6 and 8 months (panel C), stratified by age group

Page 19, lines 339-340:

and between subjects above and below median age.

Finally, this is still my major concern from the previous round: “line 211: regarding the predictive accuracy - this part is a bit challenging, using data that was used to derive formula for prediction, as a comparison and for the evaluation of its accuracy doesn’t sound quite solid. It seems quite reasonable that in this situation and on the identical population, the prediction would fit as good as it does here but this doesn’t allow for prediction in other population? This is a particular issue since there are several publications using more sophisticated statistical techniques and are still arguing this. Please consult the following reading doi.org/10.1371/journal.pone.0183250 and perhaps reconsider presenting this analysis.” the Authors provided RESPONSE: “Thank you for the thoughtful comments. We read the paper that you mentioned, and found that the content was not easy to understand. Anyway, we incorporated VEcv. We think that Lin’s concordance correlation coefficient (CCC) is another metric to quantify accuracy; thus, we also included Lin’s CCC in this revised manuscript.” Additionally, I agree with the comment 5 from the Reviewer #1 that said “a comparison of the equations obtained applying the same method described in this study to other published cohorts will be useful to further support the accuracy of the predictions for healthy individuals of different ethnicities.” In fact, the S1 Fig shows good correlation between predicted and measured values, which is expected when using the same dataset, as I previously explained. I feel this is another limitation of the study. The problem is not comparison to other ethnicity as listed in the lines 358-360 but the methodology (data) used for accuracy evaluation.

Thank you for these important comments. We agree with your comments, and described these points in the study limitation.

Page 21, lines 382-386:

Third, although we generated simple equations, a good correlation between predicted and measured values was expected because we used the same dataset. A comparison of the equations obtained applying the same method described in this study to other cohorts will be useful to further support the accuracy of the predictions for healthy individuals of different ethnicities.

Results:

Table 1: please provide appropriate labeling of the Variables i.e. Age (years), median (IQR); Male, n (%), etc. What represents the Part-time occupation? it’s unclear, it doesn’t seem a position on which they work? should be explained or modified. I don’t feel “Nucleocapsid IgG at 3/6/8 months” values should be listed here in Table 1 and are a bit misleading since these are not characteristics of the participants but something that was measured in this study and also only 3 participants were positive on this test so median of 0.02 doesn’t tell much since many had value of 0?

Thank you for these good suggestions. We incorporated them in table 1. “Part-time” means part-time worker, but it changed it to “Others”. We omitted nucleocapsid IgG values in table 1 as suggested.

Correlation coefficients (r) should be accompanied by corresponding p-values (lines 164, 165, etc.)

We incorporated p-values.

Page 9, lines 168-169:

months (r=0.86, p<0.001) and between 3 and 8 months (r = 0.82, p<0.001). The correlation was almost perfect between 6 and 8 months (r = 0.98, p<0.001).

As per propositions of PLOS One, tables should be placed right after being first mentioned in the text (Table2).

We did.

I find information from the nomograms presented in Table 4 (and 5) to be very useful. But, how was the formula used for the Predicted elapsed month = 37.5 * (1 – (2.71 / log-transformed spike IgG at 3 month)) derived? There is no explanation on this?

We added these sentences to describe how to obtain this equation in the foot note of Table 5.

Page 15, lines 266-270:

Values of the second raw of the table was calculated as follows.

From two formulae (log spike IgG at 6 months = 0.92 X log spike IgG at 3 months – 0.23, log spike IgG at 8 months = 0.86 X log spike IgG at 3 months – 0.18),

We supposed that the coefficient declined linearly according to month (M) (0.08/3 = 0.027, 0.14/5 = 0.028). 

Thus, log spike IgG of 300 = (1 – 0.027 M) x log spike IgG at 3 months – 0.23.

Predicted elapsed month (M) = 37 x (1 – (2.71 / log spike IgG at 3 month))

Discussion:

I would be more satisfied if the authors used some of the studies listed in lines 285-287 “Several studies have measured short- [28, 29] and long-term changes [4-14] in spike IgG and neutralizing antibody levels up to 9 months after two doses of BNT162b2, and all studies have revealed that antibody levels drop over time.” to make more vivid discussion, i.e., to compare their results with those obtained by previous studies (since there are many available nowadays). 

Thank you for the good suggestions. We made Table 6 in which we compared spike IgG levels at 3 and 6 months after two-dose BNT162b2 vaccination and their percent reductions, and discussed the observed results.

Page 17-19, lines 311-325:

Table 6 summarizes reported spike IgG levels at 3 and 6 months after two doses of the BNT162b2 vaccine.

Table 6: Reported spike IgG levels at 3 months and 6 months after two doses of BNT162b2 vaccine

Author Study type Subjects Country Spike IgG at 3M Spike IgG at 6M % reduction

Bayart longitudinal HCWs Belgium 6,050 AU/mL (n = 158) 949 AU/mL (n= 158) 68%

Naaber longitudinal HVs Estonia 5,226 AU/mL (n = 122) 1,383 AU/mL (n = 122) 74%

Rode longitudinal HCWs Croatia 2,977 AU/mL (n = 405*) 966 AU/mL (n = 337*) 68%

Guiomar cross-sectional HCWs Portugal 6,812 AU/mL (n =32) 1,070 AU/mL (n= 72) 86%

Israel cross-sectional PB Israel 2,383 AU/mL (n = 200) 765 AU/mL (n = 440) 68%

Kertes cross-sectional PB Israel 2,706 AU/mL (n = 827) 1,411 AU/mL (n = 1,820)** 48%

Our study longitudinal HCWs Japan 2,882 AU/mL (n= 251) 875 AU/mL (n = 250) 70%

Spike IgG levels are presented as mean or median.

HCWs, health care workers; HV, healthy volunteers; n, number; M, month; PB, population based. 

*: exact number not available

**: ≥150 days

IgG levels at 3 and 6 months varied widely among studies. The percent reduction of spike IgG levels from 3 months to 6 months ranged from 48% to 86%. However, it is interesting to note that the percent reduction of spike IgG levels in our study (70%) was quite similar to the corresponding value observed in three longitudinal studies (68%, 74%, 68%), which suggests that the decay kinetics of spike IgG after two-dose BNT162b2 vaccination may be relatively constant.

Also, the results seem quite valuable but in the discussion I miss the answer on the major question “what might be the utility of these findings” and stronger statement on why these results are valuable for practice?

line 339 seems a bit misleading, these are more Public health implication rather than Clinical implications? this section should answer the question: What would these findings add if being implemented in practice?

We modified the sentences in Clinical implication to try to answer these questions.

Page 20-21, lines 364-376:

According to the table, spike IgG levels of 900 AU/mL represent approximately the minimum amount of spike IgG necessary at 3 months to maintain spike IgG of ≥ 300 AU/mL at 6 months. The table also provides minimal spike IgG levels at 3 months required to maintain spike IgG levels of 500, 750, 1,000, 1,500 and 3,000 AU/mL at 6 or 8 months. This table provides predictive spike IgG levels at 6 months in individual subjects, for whom we have spike IgG levels at 3 months after two-doses of BNT162b2 vaccine. Since spike IgG levels at 3 months vary widely among subjects, this information is quite useful for individualized third-dose vaccine schedules. Another nomogram (Table 5) shows at which month after vaccination one expects to reach the pre-defined spike IgG cut-off level (300 AU/mL) based on levels at 3 months. This nomogram provides information when spike IgG levels become < 300 AU/mL in each subject who measured spike IgG levels at 3 months; thus, it supports individual scheduling for a third dose.

We added six references to this revised manuscript. (#13, 14, 25, 27, 28, and 29).

---

## [Editor Report · Decision Letter 2]

7 Jun 2022

Temporal changes in spike IgG levels after two doses of BNT162b2 vaccine in Japanese healthcare workers: Do spike IgG levels at 3 months predict levels 6 or 8 months after vaccination?

PONE-D-22-01729R2

Dear Dr. Takeuchi,

We’re pleased to inform you that your manuscript has been judged scientifically suitable for publication and will be formally accepted for publication once it meets all outstanding technical requirements.

Kind regards,

Etsuro Ito

Academic Editor

PLOS ONE

---

## [Editor Report · Acceptance letter]

10 Jun 2022

PONE-D-22-01729R2 

Temporal changes in spike IgG levels after two doses of BNT162b2 vaccine in Japanese healthcare workers: Do spike IgG levels at 3 months predict levels 6 or 8 months after vaccination? 

Dear Dr. Takeuchi:

I'm pleased to inform you that your manuscript has been deemed suitable for publication in PLOS ONE. Congratulations! Your manuscript is now with our production department. 

Kind regards, 

on behalf of

Prof. Etsuro Ito 

Academic Editor

PLOS ONE